# The Crazy Ovary

**DOI:** 10.3390/genes12060928

**Published:** 2021-06-18

**Authors:** Philippe Monget, Ken McNatty, Danielle Monniaux

**Affiliations:** 1UMR INRAE-CNRS-IFCE-Université de Tours, 37380 Nouzilly, France; danielle.monniaux@gmail.com; 2School of Biological Sciences, Victoria University of Wellington, Wellington 6140, New Zealand; Kenneth.mcnatty@vuw.ac.nz

**Keywords:** ovary, folliculogenesis, reactivation, primordial follicles, ovulation rate, unexpected, crazy, steroidogenesis

## Abstract

From fetal life until senescence, the ovary is an extremely active tissue undergoing continuous structural and functional changes. These ever-changing events are best summarized by a quotation attributed to Plato when describing motion in space and time—‘nothing ever is but is always becoming…’. With respect to the ovary, these changes include, at the beginning, the processes of follicular formation and thereafter those of follicular growth and atresia, steroidogenesis, oocyte maturation, and decisions relating to the number of mature oocytes that are ovulated for fertilization and the role of the corpus luteum. The aims of this review are to offer some examples of these complex and hitherto unknown processes. The ones herein have been elucidated from studies undertaken in vitro or from normal in vivo events, natural genetic mutations or after experimental inactivation of gene function. Specifically, this review offers insights concerning the initiation of follicular growth, pathologies relating to poly-ovular follicles, the consequences of premature loss of germ cells or oocytes loss, the roles of *AMH* (anti-Müllerian hormone) and *BMP* (bone morphogenetic protein) genes in regulating follicular growth and ovulation rate together with species differences in maintaining luteal function during pregnancy. Collectively, the evidence suggests that the oocyte is a key organizer of normal ovarian function. It has been shown to influence the phenotype of the adjacent somatic cells, the growth and maturation of the follicle, and to determine the ovulation rate. When germ cells or oocytes are lost prematurely, the ovary becomes disorganized and a wide range of pathologies may arise.

## 1. Introduction

The mammalian ovary has been, and continues to be, an extensively studied organ. With respect to human health and well-being, significant challenges remain, including the restoration of fertility in women with polycystic syndrome; assisting those experiencing an early depletion of their oocyte reserve and; for both human and animal conservation, new technologies to promote ovarian follicular growth in vitro in order to obtain mature oocyte for fertilization and a successful pregnancy. Many reviews have already been written on all these aspects. Of interest is the fact that mammalian ovaries sometimes exhibit quite remarkable functional quirks, whether these are part of normal physiology or observed in nature by mutations or after experiments inducing functional modifications by inactivating or enhancing gene function. The purpose of this review is to offer a few examples of the above to show that the ovary is a complex tissue to study with sometimes unpredictable (“crazy?”) results. The presented examples address: the initiation of follicular growth; the consequences of oocyte loss for somatic ovarian cells; pathologies relating to poly-ovular follicles; the roles of *AMH* (anti-Müllerian hormone) and *BMP* (bone morphogenetic protein) genes in regulating follicular growth and ovulation rate, and; some species differences in maintaining luteal function during pregnancy. The intention of this review is not to be exhaustive, but to go beyond the scope of classic reviews.

### 1.1. The Initiation of Follicular Growth: The Hippo Signaling System

Ovarian follicular formation takes place entirely within the ovigerous cords [1]. Primordial follicles are formed sequentially from the base of the cords as each oocyte and single layer of epithelial cells separate by isolating themselves by a basement membrane. Thereafter, the first follicles that form are the first to grow and this occurs even before the full complement of primordial follicles have assembled [1]. In a classic series of studies, Peters et al. showed that follicles grow sequentially and continue to grow until they die or ovulate [2]. Moreover, this process is not interrupted from the time follicles are formed and continues throughout life, irrespective of reproductive status until death or until the primordial pool is depleted. It is not known whether the order at which follicular growth is initiated is determined by the order in which the follicles are formed. However, even if there is an ordered process, it can be disrupted. In 2013, Aaron Hsueh and colleagues published an important paper showing that when segments of ovaries (i.e., ovarian fragmentation) are removed into culture, the initiation of primordial follicular growth can occur through the polymerization of actin and disruption of the Hippo signaling pathway in the ovarian cortex [3]. This disruption causes the nuclear translocation of Yes-associated protein (YAP), which interacts with transcriptional-enhanced associated domain (TEAD) transcriptional factors. In turn, this leads to the expression of CCN (cystein-rich 61, connective tissue growth factor and nephroblastoma overexpressed) growth factors as well as BIRC (baculoviral inhibitors of apoptosis repeat containing) apoptosis inhibitors. This growth activation can be enhanced further by utilizing PI3K/AKT activators that act synergistically with the inhibition of Hippo pathway.

This In Vitro Activation (IVA) approach has led to the delivery of a live offspring. Thus, it offers a practical solution for some patients who suffer from a premature ovarian insufficiency (POI). This IVA technology has also been used successfully in both Japan and China to treat POI. However, IVA is not a practical solution for more than 50% of POI patients who have few, if any, residual follicles in their ovaries (Figure 1). 

### 1.2. The Deletion of Omgc1 in Oocytes and Cyclic Ovarian Steroidogenesis in the Interstitium?

The ovum mutant candidate gene 1 (*Omcg1*) encodes a nuclear zinc finger protein involved in pre-mRNA processing including splicing and transcription-coupled repair. Oocyte-specific inactivation of *Omcg1* in early growing oocytes leads to reduced transcription, accumulation of DNA double-strand breaks as well as p63-dependent oocyte death. Mutant adult female mice are sterile with follicular growth arrested at the primary stage [4]. Remarkably, these mutant females, although sterile, exhibit vaginal cyclicity, with a normal onset of puberty and even sexual receptivity, suggesting that there is a cyclical production of oestradiol. The evidence indicates that the *Omcg1^ocKO^* ovaries are dramatically remodeled with oocyte-depleted follicles able to express the main steroidogenic enzymes through an apparent cooperation between interstitial cells and granulosa cells (Figure 2). These results are the first showing the possibility of an uncoupling between the cyclicity of sexual hormone secretion, and the follicular growth from small to preovulatory follicles. They also raise the possibility of internal clock genes within the somatic cells of the ovary. However, an alternative explanation might be a ‘cyclic-like’ positive-negative ovarian-pituitary feedback loop, but this requires further investigation. 

### 1.3. Depletion of Oocytes before Puberty and the Fate of Granulosa Cells

During the assembly of primordial follicles, significant numbers of the granulosa cells are recruited from the ovarian surface epithelium [1,5]. After the follicles have assembled, no new follicles form during the prepubertal or adult life. Therefore, the long-term viability of oocytes within primordial follicles is critical for reproductive success in later life. Although oocytes are equipped with an impressive array of repair mechanisms, their viability can be threatened by genetic mutations, X-irradiation, and chemo-toxic drugs [6,7,8]. In homozygous genetically-mutant mammals (e.g., *Dazl*; *Fancd2*; growth differentiation factor 9 *(GDF9)*; bone morphogenetic protein 15 *(BMP15)*) where germ cells or oocytes die without significant loss of the granulosa cell population [9], ovaries undergo significant, seemingly disorganized, restructuring with a diverse array of abnormal structures: these can include the formation of intraovarian cords, ducts, clusters of endocrinologically-active cells and/or variable-sized internal or surface-visible solid tissues. 

In *Dazl (−/−)* mice, devoid of the RNA-binding protein, oogenesis and follicular assembly is compromised. Consequently, follicles do not form, and the resulting somatic cell abnormalities may display variable levels of endocrine activity reminiscent of that for the aforementioned *Omgc1 (−/−)* mice but without evidence of puberty onset.

In the *GDF9* and *BMP15* mutant mice and sheep, normal populations of primordial follicles are assembled. However, after follicular growth is initiated and the epithelial-like granulosa cells become cuboidal, the absence of the oocyte-derived growth factors, GDF9 and/or BMP15, leads to the oocyte continuing to enlarge but without a concomitant proliferation in the numbers of granulosa cells [10,11]. Subsequently, the oocytes die, leaving nodules of partially differentiated granulosa cells. The nodules are sometimes observed in clusters together with solid luteal-like bodies and large cysts often with endocrine activities (e.g., oestradiol, inhibin, progesterone). 

The FANCD2 protein is normally expressed in both oocytes and granulosa cells and plays an important role in DNA repair and meiotic maturation. However, in the *Fancd2*-*KO* mice, some primordial follicles assemble around birth. All oocytes, but not the granulosa cells, die within the first three months of life. Subsequently, extensive intraovarian remodeling occurs with sex cords and epithelial tubules becoming increasingly prominent with age. Thereafter, tumors with multiple phenotypes develop, with some fulfilling the criteria of malignant carcinomas (Figure 3) [9]. One interpretation of this phenomenon is that the epithelial-like granulosa cells have reverted to their fetal phenotype and form cords in an attempt to locate germ cells.

Collectively, these data suggest that the epithelial-derived granulosa cells associated with oogonia or with oocytes in primordial follicles retain a high level of plasticity and have the potential to form ovarian cancers if the viability of the oocytes is at risk. However, once primordial follicles start to grow and the epithelial-derived follicular cells develop a granulosa cell phenotype, extensive intraovarian modeling still occurs but without the appearance of intraovarian sex cords and/or epithelial-tubules.

### 1.4. Poly-Ovular Follicles: From Physiology to Environmentally Induced Pathology

Although a single oocyte per follicle is generally the rule in mammals as in other vertebrates, poly-ovular follicles are not uncommon in the ovaries of marsupials and various eutherian mammals including humans. In the opossum ovary, some follicles can contain up to a 100 oocytes per follicle, most of them degenerating by atresia during follicular growth [12]. The percentage of poly-ovular follicles is quite variable between species and between individuals within species. It is particularly high in the ovaries of cats (up to 4% of follicles) and dogs (up to 14% of follicles), each poly-ovular follicle containing between 2 and 17 oocytes [13,14]. In humans, poly-ovular follicles and polynuclear oocytes are found in 98% of 18- to 52-year-old women and represent between 0.06% and 2.44% of the total follicular population of the ovary [15]. In poly-ovular follicles, the quality of the oocytes is often heterogeneous. Some poly-ovular follicles are able to ovulate but generally, at most a single oocyte is fertilized [14]. 

Three possible modes of formation of poly-ovular follicles have been suggested: the subsequent division of polynuclear oocytes, the fusion of previously separate follicles, and the initial failure of separation of oocytes by follicular cells in germ cell cysts during follicular formation. The first hypothesis is documented in humans with the observation of oocytes containing two germinal vesicles (0.4% of growing follicles, from [15]) and with the recovery and in vitro fertilization of bin-ovular complexes, in which the two oocytes are included in the same zona pellucida [16,17,18]. Concerning the second hypothesis, a few morphological studies in humans have suggested that bin-ovular follicles might be formed by the fusion of two nearby primordial or primary follicles [19,20]. Whatever the origin of the bin-ovular complexes, the successful fertilization of both oocytes and the development of dizygotic twins was an exception [21].

Abnormal germ-cell cyst breakdown and assembly of individual oocytes with pre-granulosa cells in the ovigerous cords of the fetal or neonatal ovary is the most likely hypothesis to explain the formation of most poly-ovular follicles. Intricate regulation of gene expression, including the oocyte-specific Figla [22], Nobox [23], and Taf4b [24] transcription factors, is critical for these processes in the mouse. The oocyte-specific secreted factors BMP15 and GDF9 might participate as well in controlling follicular formation, since the formation of poly-ovular follicles has been observed in *Bmp15−/− Gdf9+/−* mice [25]. In humans and sheep, BMP15 is not expressed until the follicle is formed and begins to grow, but it can be speculated that BMP15 secreted by the first growing follicles may activate BMPR1B (bone morphogenetic protein receptor type 1B) signaling in the ovigerous cords and influence the subsequent formation of follicles. In the mouse, somatic cell-derived growth factors such as activin/follistatin and neurotrophins also influence cyst breakdown [26,27,28], and the disintegrin Adam10 has been shown to govern the recruitment of the pre-granulosa cells in cysts [29]. The actions of these various factors converge to modulate the Jagged/Notch signaling pathway in germ cell cysts. Particularly, the *Notch2* gene encoding a Notch receptor in pre-granulosa cells orchestrates cyst breakdown, postnatal apoptosis of oocytes and primordial follicle assembly [30] and the oocyte-specific secretory proteins Jagged1 (a Notch ligand), Gdf9 and Bmp15 activate the Notch signaling pathway in pre-granulosa cells [31]. All these observations highlight the importance of crosstalk between germ cells and pre-granulosa cells for the formation of primordial follicles [32] (Figure 4). They suggest that the formation of poly-ovular follicles could result from local disequilibrium of the concentration of factors involved in this crosstalk in the fetal or neonatal ovaries of mammals.

The fetal gonad synthesizes steroids and intra-ovarian oestradiol likely participates in the physiological mechanisms regulating follicle assembly [32]. Oestrogens can control follicular formation through mechanisms involving members of the TGF-β family such as activin (monkey: [33]) and BMP2 (hamster: [34]). Whether between-species differences in intra-ovarian concentrations at the time of follicular formation can explain differences in the percentages of poly-ovular follicles observed between species and between individuals within species remains hypothetical, however. As first observed in Florida alligators [35], pharmaceutical and environmental oestrogens, such as phytoestrogens or pesticides with estrogenic action, can induce the formation of poly-ovular follicles in the ovaries of mammals. Particularly, early-life exposure to synthetic oestrogens (diethylstilbesterol) or environmental pollutants with oestrogen-like activity (genistein, zearalonone, diethylhexylphthalate) have been shown to impair the assembly of primordial follicles [36,37,38,39]. The presence of multiple poly-ovular follicles in mouse ovaries exposed to diethylhexylphthalate is associated with a decrease in the gene and protein expression of the oestradiol receptor Esr2 and components of Notch signaling [39]. Thus, deregulation of the mechanisms of germ cell cyst breakdown and assembly of individual oocytes with pre-granulosa cells in the ovigerous cords by oestrogen compounds results in the establishment of a pathological ovarian reserve of poly-ovular primordial follicles. 

### 1.5. Differences in Ovarian Follicular Characteristics between Species

#### 1.5.1. From Teleost to Tetrapod

The female reproductive system is different in several ways between teleost fish and tetrapod. In teleost (i.e., in most of them), fertilization is external, whereas in tetrapod it is internal with the exception of amphibians. This is most likely the reason why the genomes of tetrapods, and specifically mammals, have lost several genes encoding the proteins involved in sperm/egg interactions [40], including the ZP (zona pellucida) [41]. Indeed, these proteins constitute a species barrier important in water where gametes of several species are susceptible to being mixed, this barrier being less necessary when fertilization is internal. 

Secondly, most teleost ovulate tens or even hundreds of thousands of oocytes per ovulation. It is most likely for this evolutionary reason (a sort of evolutionary residue?) that mammalian ovaries still constitute a stock of several thousand oogonia during fetal life, this stock disappearing largely before puberty. However, at the time of ovulation, mammalian ovaries ovulate significantly fewer oocytes than teleost, from 1 in women, cows and mares, between 2 and 5 to 6 in ewes (see below), and between 15 and 25 in sows.

#### 1.5.2. Between Pig and Cattle

Porcine ovaries contain a large number of growing antral follicles compared to those of cattle, sheep, women, and even dogs, cats, and rodents. The ovulation rate of sows can exceed 20, with some ovulating more than 50 follicles per cycle. Indeed, the surface of the sow’s ovary looks similar to a bunch of grapes. In part, this is due to the fact that the promoter of the porcine *AMH* gene is much less active than the bovine one, with the antral follicles of the pig secreting very low levels of the anti-Müllerian hormone (AMH) compared to those of the cow. Indeed, in mammalian females, AMH plays an inhibitory role in the recruitment of primordial ovarian follicles, the growth of small preantral follicles, and in modulating the sensitivity of antral follicles to FSH (follicle stimulating hormone), thereby restricting preovulatory follicular maturation [42,43,44]. In the pig, the low levels of AMH have very little effect in inhibiting the recruitment of primordial follicles, thereby leading to a larger number of growing follicles and a higher ovulation rate, compared to that in the cow (Figure 5) [45].

#### 1.5.3. Between Different Sheep Breeds: The Role of the BMP Family Influencing Ovulation Rate

Mammals have evolved a number of strategies to limit the number of ovulated oocytes. Members of the BMP growth factor family have been identified as crucial factors in regulating ovulation rate in mammals [46] This role was revealed by the identification of several loss of function mutations in the oocyte-specific expression of GDF9 and especially BMP15 and the BMPR1B receptor [47,48]. Ewes carrying null-alleles of *GDF9* or *BMP15* in the heterozygous state have increased ovulation rates, whereas homozygous mutants are sterile. In contrast, both heterozygous and homozygous ewes, with a partial loss of function in the *BMPR1B* gene, hyper-ovulate (3 to 10 oocytes ovulated per cycle, compared to 1 or 2 in the wild-type). Another gene located on X chromosome (different from the *BMP15* gene), yet to be identified, has also been shown to be in involved in an increase in ovulation rate when it is mutated at both the homozygous and heterozygous state [49]. Surprisingly, mutations in the three *BMP15*, *BMPRIB* and the unknown X-linked genes are additive, ewes carrying the three alleles at the heterozygous state exhibiting ovulation rates >12 [50].

In most of these models, the preovulatory follicles are smaller in diameter in “mutants” compared with wild-type ewes. In most mammals, (e.g., human, sheep, pig, deer, rodents) both BMP15 and GDF9 are synthesized exclusively from oocytes. However, a notable exception is the pig, where GDF9 is also synthesized in granulosa cells [51]. Unexpectedly, it turns out that it is the oocytes and not the gonadotrophins that have a major influence on the species differences in the ovulation rate. In contrast to BMP15 and GDF9, the *BMPR1B* gene is expressed in both oocytes and the somatic cells within the ovary. It seems that the BMP system, and the relative levels of BMP15 and GDF9, determine the stage at which follicular maturation occurs. Moreover, as all the identified mutations are complete or have a partial loss of function, one can consider these mutations as having “brake-release” functions, thereby resulting in follicles ready to ovulate at smaller diameters. Despite these follicles ovulating at smaller diameters, it remains to be determined whether follicular growth occurs at a different rate. Importantly, these mutations have no consequences on the capacity of oestradiol secretion by granulosa cells ([52] for Booroola mutants [53] and the heterozygous *BMP15* mutant), we can hypothesize that the increase in the ovulation rate requires a sufficient number of granulosa cells to secrete enough oestradiol necessary for the induction of the GnRH surge [54] (Figure 6). Moreover, after ovulation, the number of steroidogenic luteal cells was found to be similar between the Booroola mutants, which is consistent with the finding that there were no genotype differences in progesterone secretion [55]. It is worth noting that partial inhibition of BMP15 and/or GDF9 bioactivity in cattle [56] can also influence the ovulation rate and that GDF9 mutations in humans have been associated with dizygotic twinning [57].

### 1.6. Why Are There So Many Luteotrophic Factors in Mammals?

Most of the elements involved in ovarian function in mammals have arisen in vertebrates, (e.g., pituitary gonadotropins and their receptors), and the follicular stages of growth are more or less conserved in mammals. However, a molecular mechanism involving mammalian ovaries is radically different between species, namely the luteotrophic factors secreted by the young embryo at the start of gestation. These factors are critical for preventing the corpus luteum from disappearing by apoptosis, which would be detrimental to the survival of the embryo. These factors are:the human chorionic gonadotrophin (hCG) in women, which binds to the LH receptor.the trophoblastin (IFNτ) in ruminants, which impairs the secretion of the PGF2α luteolysis factor by the endometrium.the different forms of prolactin in rodents, which inhibit the expression of the progesterone metabolizing 20-α-hydroxysteroid dehydrogenase (20αHSD) (Figure 7, [58]).

This evolutionary incongruity is in reality most likely not due to the ovary itself, but to the placenta, which is an organ that has evolved rather rapidly between mammals. It is very probable that the ovarian corpus luteum, in order to ensure its maintenance two weeks after ovulation in large mammals, expresses the receptor adapted to the luteotrophic factors expressed by the placenta of the species in question.

## 2. Conclusions

In the same way that it is difficult to define a border between madness and normality in the field of human psychiatry, the definition of a normal ovary is puzzling. The norm in mammals is that the adult ovary contains multiple structural units that are follicles, where each follicle contains a single oocyte interacting with its surrounding granulosa cells. Some of the above-cited examples deviate from the norm, as for example: the existence of follicles containing multiple oocytes or conversely follicles losing their oocytes prematurely with granulosa cells synthesizing and secreting steroids in a cyclic manner and/or forming ovarian tumors. The norm is also that most of the quiescent primordial follicles forming the ovarian reserve in the ovarian cortex will degenerate throughout the life; the possibility to activate them after decades of quiescence by disrupting the Hippo signaling pathway gives them an unexpected opportunity to develop. Concerning ovarian endocrinology, a dogma was also developed to suggest that the ovulation rate and the follicular size at ovulation are fixed for each animal species and depended on a dialogue existing between ovarian steroids and inhibin secreted by growing follicles and the pituitary hormones, FSH and LH. The subsequent discovery and elucidation of the roles that the ovarian-derived factors such as AMH and the BMP family play in regulating both the number and the size of the ovulating follicles are new insights, which significantly modify the aforementioned dogma. Significant questions still remain, however. For example, why have different animal species evolved so many different strategies to maintain their corpus luteum during gestation?

As we can see, the ovary of mammals is capable of functional quirks that can be linked to genetic modifications, to natural intra-species alterations, to differences in inter-species functioning, or to environmental effects. Ovarian function can also be “jostled” by artificially stimulating the recruitment of primordial follicles that one might have thought “lost” for reproduction. These “dysfunctions” are most likely largely related to the fact that the ovary is an organ in perpetual cyclical modification and that it is this level of plasticity that opens the “doors of entry” to functional modifications.

## Figures and Tables

**Figure 1 genes-12-00928-f001:**
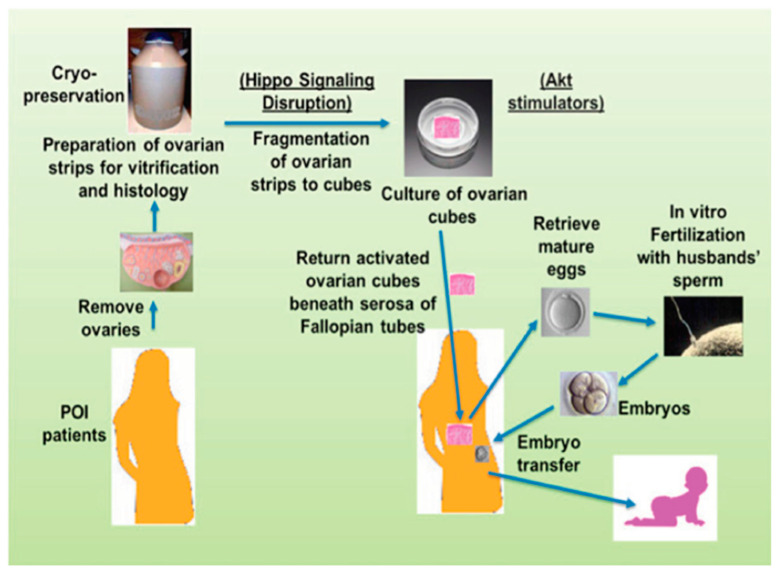
Ovarian fragmentation, in vitro Akt stimulation and auto-grafting promoted follicle growth in POI (premature ovarian insufficiency) patients and generated mature oocytes are able to be fertilized in vitro leading to a pregnancy and a baby after embryo transfer. Ovaries removed under laparoscopic were cut into strips, the latter being vitrified. After thawing, strips were fragmented into small cubes (1–2 mm^3^) that disrupted the Hippo signaling pathway, and treated with Akt stimulators (bpV (hopic) and 740YP). Forty-eight hours later, cubes were autografted under laparoscopic surgery beneath serosa of Fallopian tubes. After detection of antral follicles via transvaginal ultrasound, patients were treated with gonadotrophins, and mature oocytes were subjected to IVF (in vitro fertilization) before cryopreservation of four-cell stage embryos. Patients then received hormonal treatments to prepare the endometrium for implantation followed by transferring of thawed embryos. Several babies have been born using this technique around the world ([3], with permission).

**Figure 2 genes-12-00928-f002:**
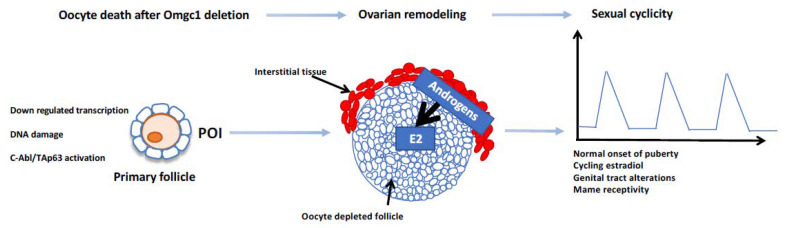
Summary of the ovarian and reproductive features of *Omcg1^ocKo^* females. Oocyte death and arrest of folliculogenesis at the primary follicle stage, leading to a kind of premature ovarian insufficiency (POI), were observed in *Omcg1^ocKo^* females. The ovarian somatic compartment was remodeled, allowing the production of oestradiol (E2), which might occur along a two-cell compartment scheme similar to what is found between theca cells and granulosa cells in preovulatory follicles. Surprisingly, despite the absence of cyclic follicular growth and then of preovulatory follicles, *Omcg1^ocKo^* females displayed features of sexual cyclicity as wild-type mice (adapted from [4]).

**Figure 3 genes-12-00928-f003:**
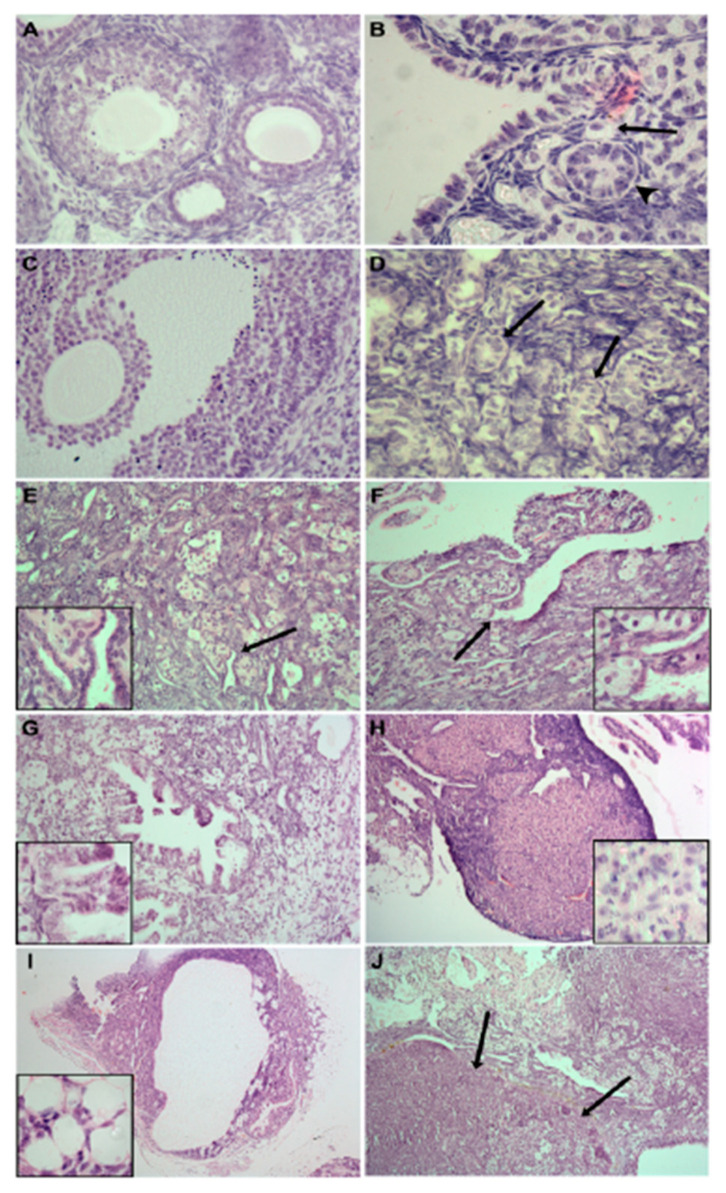
Representative micrographs of ovary sections from *Fancd2-WT* (**A**) and *Fancd2-KO* (**B**–**J**) mice at 2.5 (**A**–**C**), 5–9 (**D**–**I**) and 12 (**J**) months of age after birth. Note the appearance of primordial (arrow) (**B**), preantral (arrowhead) (**B**) and antral (**C**) follicles in ovaries of *Fancd2-KO* mice similar to that in WT mice (**A**). Common abnormal ovarian phenotypes in *Fancd2-KO* mince include: the formation of numerous sex cords (arrows) (**D**); tubules (high magnification inset, arrow) (**E**); invagination of ovarian surface epithelium (high magnification inset, arrow) (**F**); cystic papillary hyperplasia (high magnification inset, (**G**); luteomas containing large areas of cells of spongy appearance (high magnification inset, (**H**); cystadenocarcinoma whereby mitotically-active cells are invading extraovarian fat cells High magnification inset, (**I**) and; large areas containing numerous tumour phenotypes of epithelial origin, including adenocarcinomas (arrows) (**J**). Reproduced with permission from UPV/EHU Press from [9].

**Figure 4 genes-12-00928-f004:**
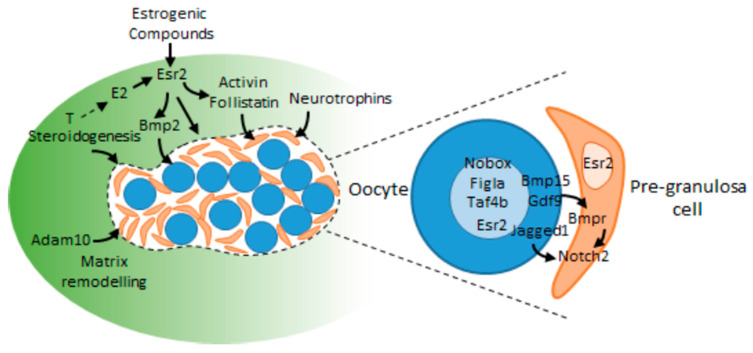
Schematic representation of some mechanisms currently known to regulate germ cell cyst breakdown and follicle assembly in the fetal or neonatal ovary. A germ cell cyst is represented (delineated by a dotted line), containing oocytes (blue round cells) and pre-granulosa cells (small pink cells). Some known interactions between germ and somatic cells participating to these processes are zoomed in the right part of the figure (adapted from [32]).

**Figure 5 genes-12-00928-f005:**
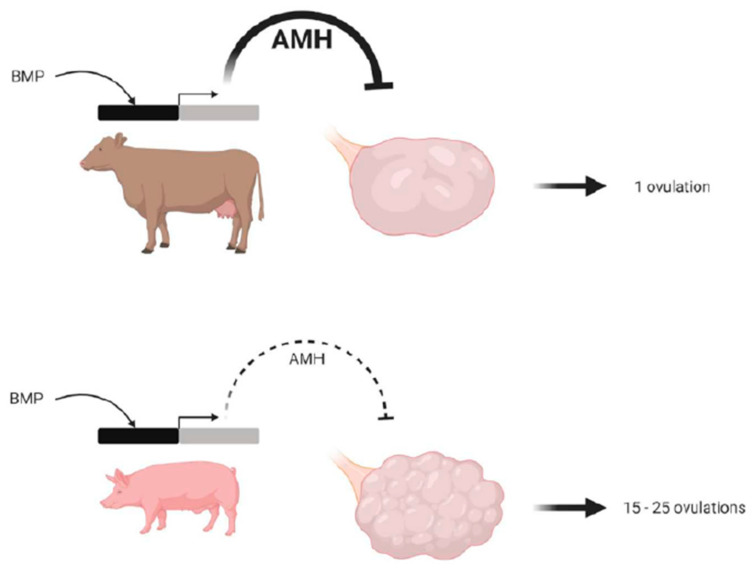
Differences between the bovine and porcine species in terms of anti-Müllerian hormone (AMH) production that could explain the regulation of ovulation number. We propose this model to explain a potential role of AMH in the regulation of the ovulation rate in porcine compared to bovine ovaries. Porcine *AMH* promoter and porcine granulosa cells are less sensitive to bone morphogenetic protein (BMP) stimulation, leading to a low production of AMH by porcine compared to bovine growing antral follicles. As AMH is known to be an inhibitor primordial follicle activation, this low level of AMH in the pig would lead to a huge number of growing follicles in the porcine compared to the bovine species. The reduced BMP sensitivity of granulosa cells and the low intra-follicular AMH concentrations of antral follicles could contribute to sensitizing granulosa cells to FSH (follicle stimulating hormone), resulting in a high follicular survival rate in the cohort of terminally developing follicles and a higher ovulation rate in the porcine, compared to the bovine species (from [45], with permission).

**Figure 6 genes-12-00928-f006:**
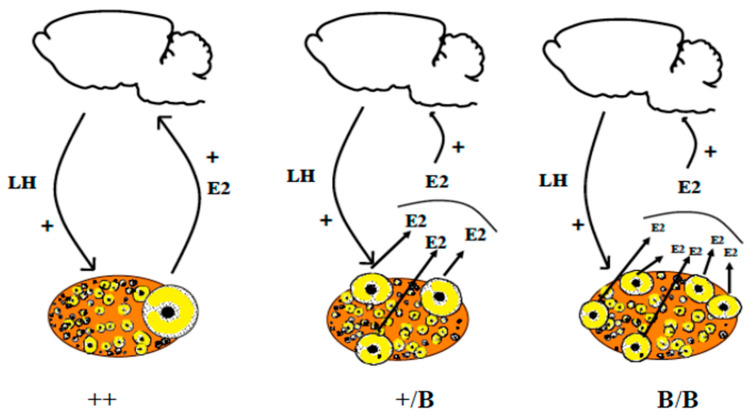
Poly-ovulation in ewes carrying loss of function mutations in the ovarian BMP system. It seems that BMP15 and GDF9 determine the stage at which follicular maturation occurs. In the case of loss of function, one can consider these mutations as having “brake-release” functions, thereby resulting in follicles ready to ovulate at smaller diameters. Additionally, as these mutations have no consequences on the capacity of oestradiol secretion by granulosa cells ([52] for Booroola mutants; [53] for heterozygous *BMP15* mutant), we can hypothesize that the increase in the ovulation rate is due to the need to have a sufficient number of granulosa cells to secrete enough oestradiol necessary for the induction of the GnRH surge [54].

**Figure 7 genes-12-00928-f007:**
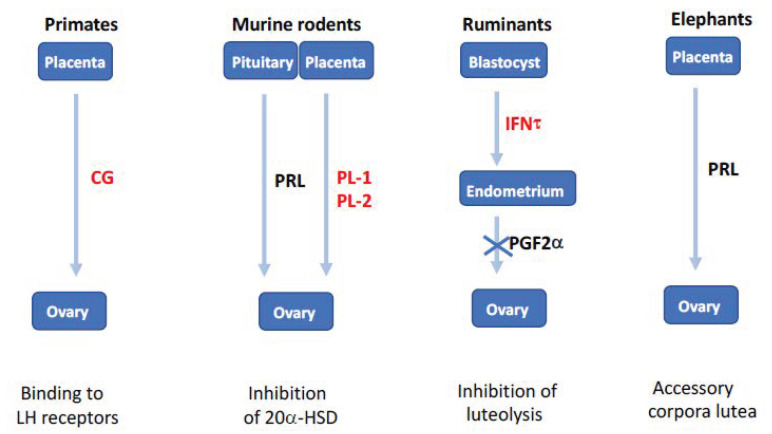
Evolution and diversity of non-steroidal luteotrophic factors in eutherians. In primates, duplication of the luteinizing hormone (*LH*) gene generated chorionic gonadotropins (CG), which are expressed by trophoblast and impair corpus luteum lysis in the first months of pregnancy. In rodents, pituitary prolactin (PRL) released in response to coitus inhibits 20-hydroxysteroid dehydrogenase (20α-HSD), this function being subsequently assumed by placental lactogens (PL-1 and PL-2). In ruminants, duplication of the *IFNW* gene generated interferon-τ (IFNT) secreted by the blastocyst and acting on the endometrium to inhibit the prostaglandin F2α (PGF2α) luteolytic signal. In elephants, the *PRL* gene expressed in the placenta is responsible for pregnancy maintenance by accessory corpora lutea. In red, proteins born after gene duplication (adapted from [58]).

## Data Availability

Not applicable.

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
