# Peer review of "The Crazy Ovary"

_genes, 2021, doi:10.3390/genes12060928_

Round 1
Reviewer 1 Report
Overall, the review is well-written and addresses an important topic. The examples presented in the review show the differences in the physiological processes of the ovary among species and how those contribute to its normal function. However, the submitted review is challenging to appreciate for a couple of reasons. The rationale for the scientific literature that was selected is difficult to grasp. The manuscript contains several interesting threads, but without a clear structure, several points raised are a little bit obscure leaving it to the reader to unpack them. The authors could significantly improve their review by providing a more defined structure to the article.
Other than that there were the following minor observations:
- Figure 1. should be re-organized. It is not immediate the flow of the figure. I see the arrows that guide the reader but there is quite a bit of up and down
- Figure 2. shares the issue of figure 1. It is difficult to read unless the reader goes to the legend
- Line 291 Table 15 should be replaced with BMP15
Author Response
Reviewer 1.
Overall, the review is well-written and addresses an important topic. The examples presented in the review show the differences in the physiological processes of the ovary among species and how those contribute to its normal function. However, the submitted review is challenging to appreciate for a couple of reasons. The rationale for the scientific literature that was selected is difficult to grasp. The manuscript contains several interesting threads, but without a clear structure, several points raised are a little bit obscure leaving it to the reader to unpack them. The authors could significantly improve their review by providing a more defined structure to the article.
à The structure of the article is announced more clearly in the revised manuscript.
Other than that there were the following minor observations:
- Figure 1. should be re-organized. It is not immediate the flow of the figure. I see the arrows that guide the reader but there is quite a bit of up and down.
à I'm a little confused, because this figure is the original figure of the paper from Kawamura et al., and I strongly prefer to put the original in this review (I have copyright permission).
- Figure 2 shares the issue of figure 1. It is difficult to read unless the reader goes to the legend.
à I have changed the thickness of the arrow between androgens and Estradiol (E2) to better distinguish it from the arrows indicating the different structures. And I have added some arrows and text to clarify this figure.
- Line 291 Table 15 should be replaced with BMP15
à I don't see any "Table 15" in the text. But we have corrected a typing mistake and replaced BMP-15 by BMP15.
Reviewer 2 Report
The manuscript is well-written and well-supported. The examples used to highlight the uniqueness of ovarian function are interesting and timely. Some suggestions to improve the manuscript are outlines below in the order in which they appear in the manuscript.
Line 14: "Unpredicted"? Perhaps unpredictable?
Line 27: Remove "the" before polycystic
Line 57: "Overexpression of nephroblastoma" It's not clear what this phrase is referring to? Is CCN61 over-expressed in nephroblastomas?
Lines 108-110: Perhaps also include a discussion of the Foxl2 mutant in this section. This mutant undergoes a female to male sex reversal when deleted in the adult.
Lines 143-145: It would be helpful in this section to more closely describe the similarities and differences among the various examples of oocyte loss and stroma reorganization. This is an area that has not been covered in depth before and could perhaps lead to identification of common features. A figure illustrating changes in the various models would better illustrate any parallels that may exist.
Line 154: Polynuclear? More information on this type of oocyte would be interesting to include. Are these oocytes with multiple GVs? or multiple nucleoli? or are these the oocyte cysts that breakdown after birth?
Line 279: As pointed out below the reason ovulation of multiple follicles occurs at a smaller size is due to the combined production of hormones from several follicles. This sentence implies an intrinsic change in the readiness of the follicle to ovulate early.
Section 1.6. This section seems underdeveloped. An entire figure is devoted to this topic and yet very little discussion is included. For example, how does 20alpha-HSD influence CL function? What does this enzyme do?
Author Response
Reviewer 2.
Comments and Suggestions for Authors
The manuscript is well-written and well-supported. The examples used to highlight the uniqueness of ovarian function are interesting and timely. Some suggestions to improve the manuscript are outlines below in the order in which they appear in the manuscript.
Line 14: "Unpredicted"? Perhaps unpredictable?
à Replaced.
Line 27: Remove "the" before polycystic
à Done.
Line 57: "Overexpression of nephroblastoma" It's not clear what this phrase is referring to? Is CCN61 over-expressed in nephroblastomas?
à It is a mistake. We have changed the sentence.
Lines 108-110: Perhaps also include a discussion of the Foxl2 mutant in this section. This mutant undergoes a female to male sex reversal when deleted in the adult.
à We prefer not to talk about Foxl2 because the consequences of its inactivation are not the same in goats and mice. In addition, in mice, the model is a classic ko, whereas in goats, the known model is a deletion of 12 kb upstream of Foxl2, responsible for the loss of expression of Foxl2 AND of at least 2 long non-coding RNAs.
Lines 143-145: It would be helpful in this section to more closely describe the similarities and differences among the various examples of oocyte loss and stroma reorganization. This is an area that has not been covered in depth before and could perhaps lead to identification of common features. A figure illustrating changes in the various models would better illustrate any parallels that may exist.
à We agree with this comment but answering this question would require another review on its own.
Line 154: Polynuclear? More information on this type of oocyte would be interesting to include. Are these oocytes with multiple GVs? or multiple nucleoli? or are these the oocyte cysts that breakdown after birth?
à These oocytes have multiple (generally only two) germinal vesicles. We have made this point clear in the revised manuscript.
Line 279: As pointed out below the reason ovulation of multiple follicles occurs at a smaller size is due to the combined production of hormones from several follicles. This sentence implies an intrinsic change in the readiness of the follicle to ovulate early.
à We have added a sentence to clarify this point.
Section 1.6. This section seems underdeveloped. An entire figure is devoted to this topic and yet very little discussion is included. For example, how does 20alpha-HSD influence CL function? What does this enzyme do?
à We have added some explanations in this section.
Round 2
Reviewer 1 Report
The authors have addressed the comments and the manuscript benefited from these edits.